# Specialized heart failure clinics versus primary care: Extended registry-based follow-up of the NorthStar trial

**Morten Malmborg**[1]*, **Ali Assad Turky Al-Kahwa**[2], **Lars Kober**[3,4], **Christian Torp-Pedersen**[5,6], **Jawad H. Butt**[3], **Deewa Zahir**[2], **Christian D. Tuxen**[7], **Mikael K. Poulsen**[8], **Christian Madelaire**[2], **Emil Fosbol**[3], **Gunnar Gislason**[1,2,4,9], **Per Hildebrandt**[10,11], **Charlotte Andersson**[12], **Finn Gustafsson**[3,4], **Morten Schou**[2]

1 Danish Heart Foundation, Copenhagen, Denmark, 2 Department of Cardiology, Herlev and Gentofte Hospital, Herlev, Hellerup, Denmark, 3 Department of Cardiology, Rigshospitalet, Copenhagen, Denmark, 4 Department of Clinical Medicine, University of Copenhagen, Copenhagen, Denmark, 5 Department of Clinical Research and Cardiology, Nordsjaellands Hospital, Hillerød, Denmark, 6 Department of Cardiology, Aalborg University Hospital, Aalborg, Denmark, 7 Department of Cardiology, Bispebjerg-Frederiksberg Hospital, Frederiksberg, DK, Denmark, 8 Department of Cardiology, Odense University Hospital, Odense, Denmark, 9 The National Institute of Public Health, University of Southern Denmark, Copenhagen, Denmark, 10 Department of Cardiology, Zealand University Hospital, Roskilde, Denmark, 11 Frederiksberg Heart Clinic, Frederiksberg, Denmark, 12 Department of Medicine, Section of Cardiovascular Medicine, Boston Medical Center, Boston University School of Medicine, Boston, MA, United States of America

* mortenmalmborg@gmail.dk

## Abstract

### Background

Whether continued follow-up in specialized heart failure (HF) clinics after optimization of guideline-directed therapy improves long-term outcomes in patients with HF with reduced ejection fraction (HFrEF) is unknown.

### Methods and results

921 medically optimized HFrEF patients enrolled in the NorthStar study were randomly assigned to follow up in a specialized HF clinic or primary care and followed for 10 years using Danish nationwide registries. The primary outcome was a composite of HF hospitalization or cardiovascular death. We further assessed the 5-year adherence to prescribed neurohormonal blockade in 5-year survivors. At enrollment, the median age was 69 years, 24,7% were females, and the median NT-proBNP was 1139 pg/ml. During a median follow-up time of 4.1 ($Q_1$-$Q_3$ 1.5–10.0) years, the primary outcome occurred in 321 patients (69.8%) randomized to follow-up in specialized HF clinics and 325 patients (70.5%) randomized to follow-up in primary care. The rate of the primary outcome, its individual components, and all-cause death did not differ between groups (primary outcome, hazard ratio 0.96 [95% CI, 0.82–1.12]; cardiovascular death, 1.00 [0.81–1.24]; HF hospitalization, 0.97 [0.82–1.14]; all-cause death, 1.00 [0.83–1.20]). In 5-year survivors (N = 660), the 5-year adherence did not differ between groups for angiotensin-converting enzyme inhibitors (p = 0.78), beta-blockers (p = 0.74), or mineralocorticoid receptor antagonists (p = 0.47).

**Data Availability Statement:** Due to restrictions related to Danish law and protecting patient privacy, the combined set of data utilized in this present study can only be made available through a

trusted third party, Statistics Denmark. Requests for data may be sent to Statistics Denmark at: http://www.dst.dk/en/OmDS/organisation/TelefonbogOrg.aspx?kontor=13&tlfbogsort=sektion or the Danish Data Protection Agency: https://www.datatilsynet.dk/english/contact-us.

**Funding:** This work was funded by the Danish Heart Foundation (Grant number: 18-R121-A8218). The funders had no role in study design, data collection and analysis, decision to publish, or preparation of the manuscript.

**Competing interests:** I have read the journal's policy and the authors of this manuscript have the following competing interests: Lars Kober: Speakers honorarium from Novo, Novartis, AstraZeneca and Boehringer. Christian Torp-Pedersen: Grants for studies from Bayer and Novo Nordisk. Finn Gustafsson: Advisor (Bayer, Abbott, Boehringer-Ingelheim, Pfizer, Alnylam, Ionis, Pharmacosmos, Amgen), speakers fee (Orion Pharma, Astra-Zeneca. Morten Schou: lecture fee Novo, Bohringer, Astra, Novo.

**Abbreviations:** ARA, Aldosterone receptor antagonist; HF, Heart failure; HFrEF, Heart failure with reduced left ventricular ejection fraction; NT-proBNP, N-terminal pro-B-type Natriuretic Peptide; HFC, Heart failure clinic; PC, Primary care; RASi, Renin-angiotensin-system inhibitors; BB, beta-blockers; MRA, mineralocorticoid receptor antagonists; CV, Cardiovascular; PDC, proportion of days covered.

## Conclusions

HFrEF patients on optimal medical therapy did not benefit from continued follow-up in a specialized HF clinic after initial optimization. Development and implementation of new monitoring strategies are needed.

## Introduction

It is well-established that in patients with heart failure (HF) with reduced left ventricular ejection fraction (HFrEF), an integrated HF clinic program including implementation of disease-modifying drugs and devices, patient education in self-care, and referral for physical training improves long-term clinical outcomes [1]. However, after the completion of such a program, it is unclear whether patients should be followed in specialized HF clinics, by their general practitioner only, or both [2]. The decision may depend on the patient's risk of deterioration and the organization of the healthcare system in the respective countries.

In The NorthStar study, patients with HFrEF receiving optimal medical therapy did not benefit from continued follow-up in a specialized HF clinic compared with follow-up in primary care during a median follow-up of 2.5 years, even in high-risk patients, defined as patients with an N-terminal pro-B-type Natriuretic Peptide (NT-proBNP) $\geq$1000 pg/ml [2]. These initial results were in accordance with a pharmacy intervention study in primary care in the United Kingdom [3] and an HF clinic intervention trial (COACH 2) from The Netherlands [4]. However, since the NorthStar study was conducted, the prognosis of HFrEF patients has improved, and long-term data evaluating the organization of care are lacking. During such a long period new drugs and devices may be developed and adherence to HF guideline therapy may decline in case of lack of continued focus on this area.

Therefore, in a *posthoc* analysis of the NorthStar study, we evaluated the effect of extended follow-up in a specialized HF clinic, compared with follow-up in the primary care, on the long-term risk of HF hospitalization and cardiovascular death, and adherence to the neurohormonal blockade, in medically optimized HFrEF patients.

## Methods

### Study design

This is an extended registry-based follow-up study of patients randomized in The NorthStar Study. The NorthStar Study was an investigator-initiated, multi-center, randomized, open-labeled, blinded endpoint trial (PROBE) [2, 5]. HFrEF patients from 18 out of 40 HF clinics (HFC) in Denmark were educated in self-care and optimized in medical therapy according to contemporary guidelines in HFC. Next, patients were randomized to either continued follow-up in a specialized HFC or discharge to primary care (PC). Patients allocated to usual care arranged an individual follow-up program with their general practitioner, whereas patients allocated to an extended follow-up in the HFC with visits at 1–3-month intervals, in which adherence to treatment, symptoms, risk factors, and comorbidities were monitored and controlled. In the trial, the median follow-up period was 2.5 years (up to 5 years of follow-up) and the intervention did not affect the primary outcome (the composite of death from any cause or admission for a protocol-specified CV cause), cardiovascular hospitalization, or death of any cause [2].

## Data sources

All residents in Denmark receive a unique and permanent civil registration number at birth or immigration that enables individual-level linkage between nationwide registries and the electronic case report from the NorthStar study. Baseline data including demographic variables, clinical biochemistry, left ventricular ejection fraction, and NT-proBNP levels were available from the trial's electronic case report form. For this extended follow-up analysis, we further obtained data from 1) the Danish Civil Registration System registry (sex, date of birth, immigration, emigration, and vital status), 2) the Danish National Patient Registry (discharge diagnoses coded according to the International Classification of Diseases (ICD)-10 since 1994), and 3) the Danish National Causes of Death Registry (date of death and underlying causes of death from death certificates). All registries have been validated previously [6–8].

## Study population

Patients with HFrEF (left ventricular ejection fraction ≤0.45) were included between 2005 to 2009 from 18 out of 40 public HF clinics in Denmark according to the inclusion and exclusion criteria (S1 Methods). Patients were educated in self-care and HF disease and were on optimal medical therapy at the study baseline. Up-titration of renin-angiotensin-system inhibitors (RASi), beta-blockers (BB), and mineralocorticoid receptor antagonists (MRA) was performed according to international guidelines, and devices were implanted if considered indicated [9]. At the time of randomization, patients with a left ventricular ejection fraction ≤ 0.45 were considered to have HFrEF in Denmark.

## Study outcomes

The primary outcome was the composite of HF hospitalization (defined as an overnight stay with one of the following hospital discharge codes for HF: ICD-10-codes I110, I130, I132, I420, I426-429, I500-503, I508-509) or cardiovascular (CV) death as the underlying cause of death (ICD-10-codes I00-99), whichever came first. The secondary outcomes were the components of the primary outcome and death of any cause. A discharge diagnosis of HF has been validated with a positive predictive value of 81% [10].

## Dose and treatment duration

The National Prescription Registry does not include information on the prescribed daily dosage of the medication, but rather the date of dispensing, strength, and quantity. Among 5-year survivors, we created an algorithm, in which a minimum, maximum, and typical daily dosage of used medication was defined for each RASi, BB, and MRA dispensed between 1 January 2004 and 31 December 2018. Based on methods described previously [11], we calculated whether patients at any time had tablets available or not. We defined a patient as receiving treatment if tablets were available.

## Statistical analyses

All individuals were followed from the date of randomization until the first event of interest, death, or emigration, whichever came first.

The baseline characteristics from Schou et al. [2] have been presented in the study (Table 1). In 5-year survivors, we estimated the prevalence of patients with an implantable cardioverter-defibrillator (ICD-10 codes BFCB) and a cardiac resynchronization therapy device (ICD-10 codes BFCA4-6, BFCA21, BFCB03, BFCB21). In the main analysis, we estimated the absolute 10-year risk of the primary and secondary endpoints using the Aalen-Johansen

**Table 1. Population characteristics at the time of randomization.**

| Characteristics | Heart failure clinic (*n* = 460) | Primary care (*n* = 460) |
|---|---|---|
| **Demographic** | | |
| Age (years, median) | 69 (47–86) | 69 (43–86) |
| Female (sex) | 106 (23) | 122 (27) |
| **Clinical** | | |
| NYHA class I–II | 411 (89) | 410 (88) |
| Blood pressure (mmHg) | | |
| Systolic | 127 (90–177) | 124 (90–166) |
| Diastolic | 75 (50–92) | 73 (50–100) |
| Heart rate (b.p.m.) | 65 (49–92) | 66 (48–95) |
| Left ventricular ejection fraction | 0.32 (0.15–0.45) | 0.30 (0.15–0.45) |
| Atrial fibrillation | 152 (33) | 146 (32) |
| LBBB in non-paced ECG | 76 (19) | 92 (23) |
| Body mass index (kg/m$^2$) | 26 (20–37) | 26 (19–40) |
| Heart failure etiology | | |
| Non-ischaemic | 188 (41) | 193 (43) |
| Ischaemic | 268 (59) | 255 (57) |
| Minnesota Living with Heart Failure Questionnaire Score | 25 (0–75) | 22 (0–73) |
| Duration of the basic HFC program, months | 9 (2–61) | 9 (2–63) |
| **Medical history** | | |
| Admission within 12 months | 188 (41) | 207 (45) |
| Hypertension | 193 (43) | 183 (40) |
| Myocardial infarction | 238 (52) | 219 (48) |
| Previous PCI/CABG | 190 (41) | 188 (41) |
| Stroke or TCI | 53 (12) | 56 (12) |
| Peripheral vascular disease | 37 (8) | 42 (9) |
| Stable angina pectoris | 36 (8) | 41 (9) |
| Diabetes | 85 (18) | 85 (18) |
| Chronic obstructive pulmonary disease | 62 (13) | 73 (16) |
| **Laboratory tests** | | |
| NT-proBNP (pg/mL) | 793 (63–6720) | 803 (58–7517) |
| Haemoglobin (mmol/L) | 8.6 (6.6–10.3) | 8.5 (6.70–10.4) |
| Anemia | 94 (21) | 102 (23) |
| Sodium (mmol/L) | 140 (131–145) | 140 (130–145) |
| Hyponatremia | 76 (17) | 72 (16) |
| Potassium (mmol/L) | 4.3 (3.5–5.1) | 4.3 (3.5–5.2) |
| Creatinine (μmol/L) | 88 (62–185) | 91 (62–193) |
| Estimated GFR$_{MDRD}$ –mL/min/1.73 m$^2$ | 69 (30–118) | 66 (30–121) |
| Estimated GFR$_{MDRD}$ ≤ 60 mL/min/1.73 m$^2$ | 152 (33) | 179 (39) |
| **Medication** | | |
| ACE-I/ARB | 392 (85) | 408 (89) |
| ACE-I/ARB, max. target dose | 306 (66) | 313 (68) |
| BB | 388 (84) | 391 (85) |
| BB, at the target dose | 243 (53) | 226 (49) |
| MRA/ARA | 143 (31) | 153 (33) |
| Loop diuretics | 259 (56) | 272 (59) |
| Furosemide doses (mg/24 h) | 40 (0–240) | 40 (0–290) |

(*Continued*)

**Table 1.** (Continued)

| Characteristics | Heart failure clinic (*n* = 460) | Primary care (*n* = 460) |
|---|---:|---:|
| Thiazide | 42 (9) | 30 (7) |
| Digoxin | 71 (15) | 66 (14) |
| Lipid-lowering agents | 296 (64) | 303 (66) |
| Antiplatelets | 322 (70) | 338 (73) |
| Anticoagulants | 142 (31) | 117 (25) |
| Nitrates | 29 (6) | 40 (9) |
| Gout medication | 35 (8) | 26 (6) |
| Antidepressants | 32 (7) | 26 (6) |
| Amiodarone | 14 (3) | 17 (4) |
| **Device therapy** | | |
| ICD | 42 (9) | 32 (7) |
| CRT | 7 (2) | 7 (2) |

Abbreviations: NYHA class, New York Heart Association; LBBB, left bundle branch block; HFC, heart failure clinic; PCI, percutaneous intervention; CABG, coronary artery bypass graft; TCI, transitory cerebral ischemia; NT-proBNP, amino-terminal-pro-brain-natriuretic-peptide; estimated $GFR_{MDRD}$, estimated glomerular filtration rate (modification of diet in renal disease formula); ACE-I, angiotensin-converting enzyme inhibitors; ARB, angiotensin receptor blocker; BB, beta-blocker; ARA, aldosterone receptor antagonist; MRA, mineralocorticoid receptor antagonists; ICD, implantable cardiac defibrillator; CRT, cardiac resynchronization therapy. [a]Anaemia was defined according to WHO: <7.4 mmol/L for females and <8.1 mmol/L for males. [hyponatremia] was defined as plasma sodium <136 mmol/L.

estimator, which incorporates competing risks of death. For the primary outcome and CV death, the competing risk was non-CV death, and all-cause death for the HF hospitalization outcome. We used a Cox regression model to calculate hazard ratios with 95% confidence intervals of the primary and secondary outcomes comparing the PC and HFC groups. In subgroup analyses, we used a Cox regression model to calculate the hazard ratio of the primary outcome comparing the PC and HFC groups in subgroups of risk factors (NT-proBNP ≥1,000/<1,000, age ≥70/<70, left ventricular ejection fraction <30/≥30, The New York Heart Association functional class 3/1-2, atrial fibrillation yes/no, estimated glomerular filtration rate <60/≥60, anemia yes/no (yes: <7.4 mmol/L for women, <8.1 mmol/L for men), hyponatremia yes/no, Minnesota score ≥25/<25, previous myocardial infarction yes/no, diabetes yes/no, use of aldosterone antagonists yes/no, use of ≥80 mg furosemide/24h yes/no).

In supplementary analyses, we included an analysis comparing groups based on the number of HF rehospitalizations. In 2-year survivors, patients were followed from 2 years following randomization until the event of interest, death, or 10 years following randomization. The absolute 8-year risk of the primary and secondary endpoints was estimated in groups defined by the number of HF hospitalizations (0, 1, or >1) within the first 2 years following randomization using the Aalen-Johansen estimator, incorporating the competing risks of death (Kaplan-Meier for all-cause death). We used a Cox regression model adjusted for numbers of prior HF hospitalizations to calculate hazard ratios with 95% confidence intervals of the primary and secondary outcomes comparing the PC and HFC groups. To analyze the main exposure in groups by the number of prior HF hospitalization, the interaction between the main exposure (PC vs HFC) and the number of prior HF hospitalizations was analyzed in the same Cox models.

Lastly, we used consecutive claimed prescriptions to calculate the drug adherence level as a proportion of days covered (PDC) for RASi, BB, and MRA among 5-year survivors. PDC was calculated for each drug group separately as days exposed to the medication within the five

years following randomization (PDC = days covered / 365.25*5). Zero one inflated beta-regression was used to calculate differences in means of PDC for each drug group between the PC and HF clinic groups.

The significance level for all analyses was set to 5% (two-sided). All statistical analyses were conducted using R version 3.6.1 [12].

### Ethics

The trial was approved by the Danish Ethics Committee (KF 01/2724936), and all the patients provided written informed consent. Retrospective register studies do not need ethical approval in Denmark. The extended register-based follow-up was approved by the Danish Data Protection Agency (Approval number P-2019-396).

### Results

A total of 6,180 patients had at least one visit to one of the HFCs during the randomization period, of which 1,640 fulfilled the criteria for stability and were eligible (S1 Fig). 54 patients were excluded due to the presence of exclusion criteria, 103 declined to participate in a scientific project, 107 wanted to be followed by their general practitioner, 256 patients had an NT-proBNP <1000 pg/mL after allocation in this group was completed, and 199 consenting patients were randomly assigned for a parallel study. The remaining 921 patients were enrolled, of which 461 were randomly assigned to undergo follow-up in the HFC, and 460 to receive usual care. The baseline characteristics of the patients were similar between the two groups (Table 1). On average, all patients had been followed already for 9 (2–63) months in the HFC at the time of randomization.

After 5 years of follow-up, 335 (72.7%) patients in the HFC group and 325 (70.7%) patients in the PC group were still alive. At 5 years following randomization, 64 (19.1%) and 62 (19.1%) patients had an implantable cardioverter-defibrillator in the HFC and PC group, respectively (chisq test p-value = 1.00), and 25 (7.5%) and 30 (9.2%) patients had a cardiac resynchronization therapy device in the HFC and PC group, respectively (chisq test p-value = 0.50).

### 10-year risks of primary and secondary outcomes

After 10 years of follow-up, CV death or an HF hospitalization occurred in 321 patients from the HFC group and 325 patients from the PC group, corresponding to a 10-year risk of 70.5% (CI 66.0–74.4) and 69.8% (CI 65.3–73.7), respectively (Fig 1). We observed no difference between the PC and HFC groups (Hazard ratio 0.96 (0.82–1.12), p-value 0.59; Fig 1). Similarly, we observed no difference in the absolute 10-year risk of all-cause death (HFC: n = 238, 51.7% [CI 46.9–56.1]; PC: n = 233, 50.7% [CI 45.9–55.1]), CV death (HFC: n = 174, 40.4% [CI 35.5–44.9]; PC: n = 171, 40.7% [CI 35.7–45.3]) and HF hospitalization (HFC: n = 297, 69.2% [CI 64.3–73.4]; PC: n = 293, 68.6% [CI 45.9–72.9]) (Fig 1). Among the most common causes of rehospitalization during follow-up, a rehospitalization for HF was the most common cause of any rehospitalization (S1 Table).

### Subgroup analyses

Patients randomized to follow-up in HF clinics did not have a significantly different risk of the primary composite outcome in any of the predefined subgroups compared with those randomized to follow-up in primary care (Fig 2). Of note, atrial fibrillation was non-significantly associated with a higher hazard rate in the HFC group compared to the PC group (HR 0.78 [0.60–1.02], p = 0.07). However, due to a low number of patients and outcomes, the non-significant result may be due to power issues.

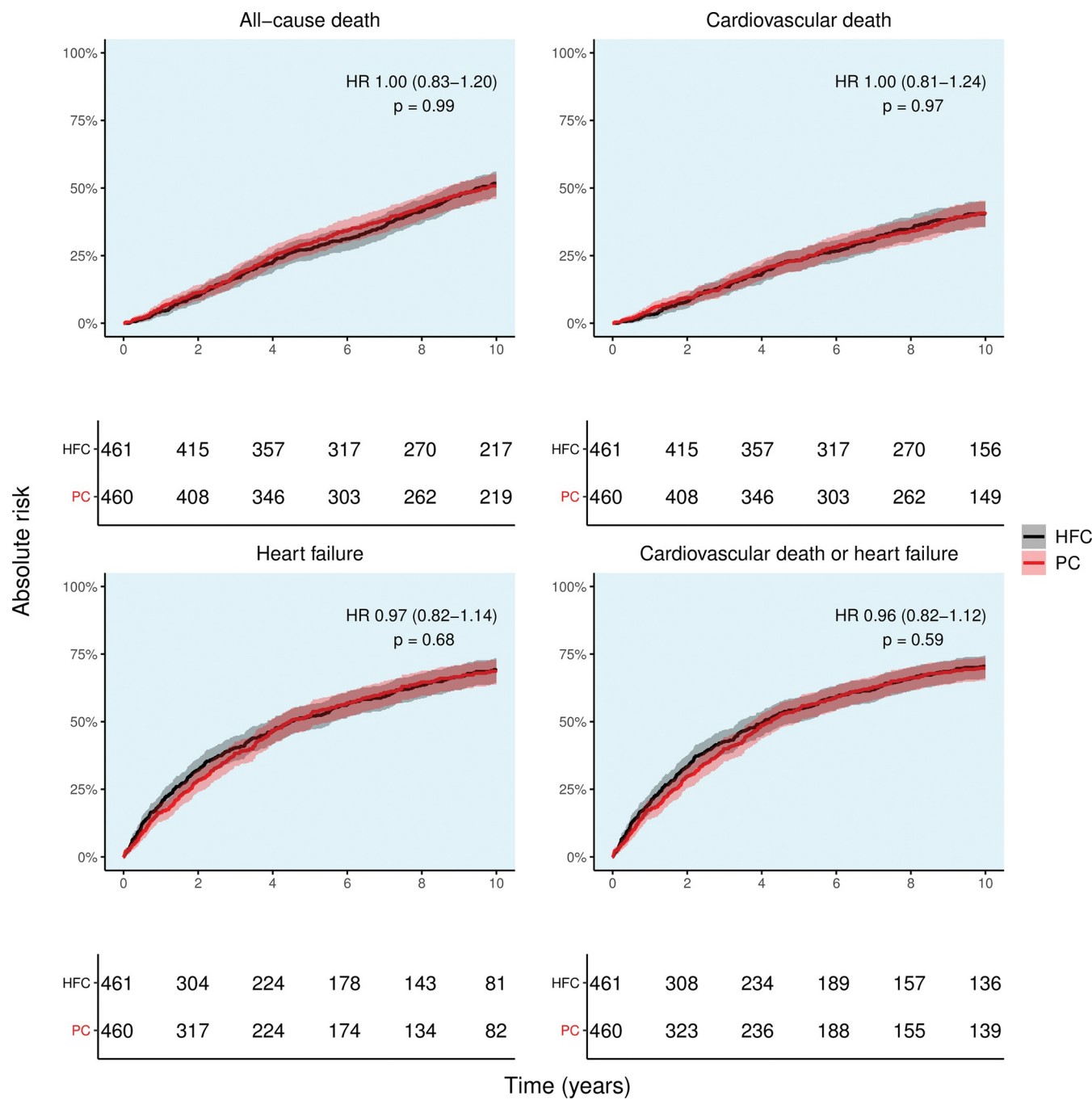

**Fig 1. 10-year absolute risk of all-cause death, cardiovascular death, discharge for heart failure, and cardiovascular death or discharge for heart failure.** Abbreviations: PC = primary care, HFC = heart failure clinic, HR = hazard ratio.

## Supplementary analyses

At 2 years following randomization, 415 (HF = 0, 304 patients; HF = 1, 64 patients; HF >1, 47 patients) and 408 patients (HF = 0, 317 patients; HF = 1, 46 patients; HF >1, 45 patients) were alive in the HFC and PC group, respectively. We observed that the number of prior HF hospitalizations was strongly associated with the primary and secondary outcomes in both the PC

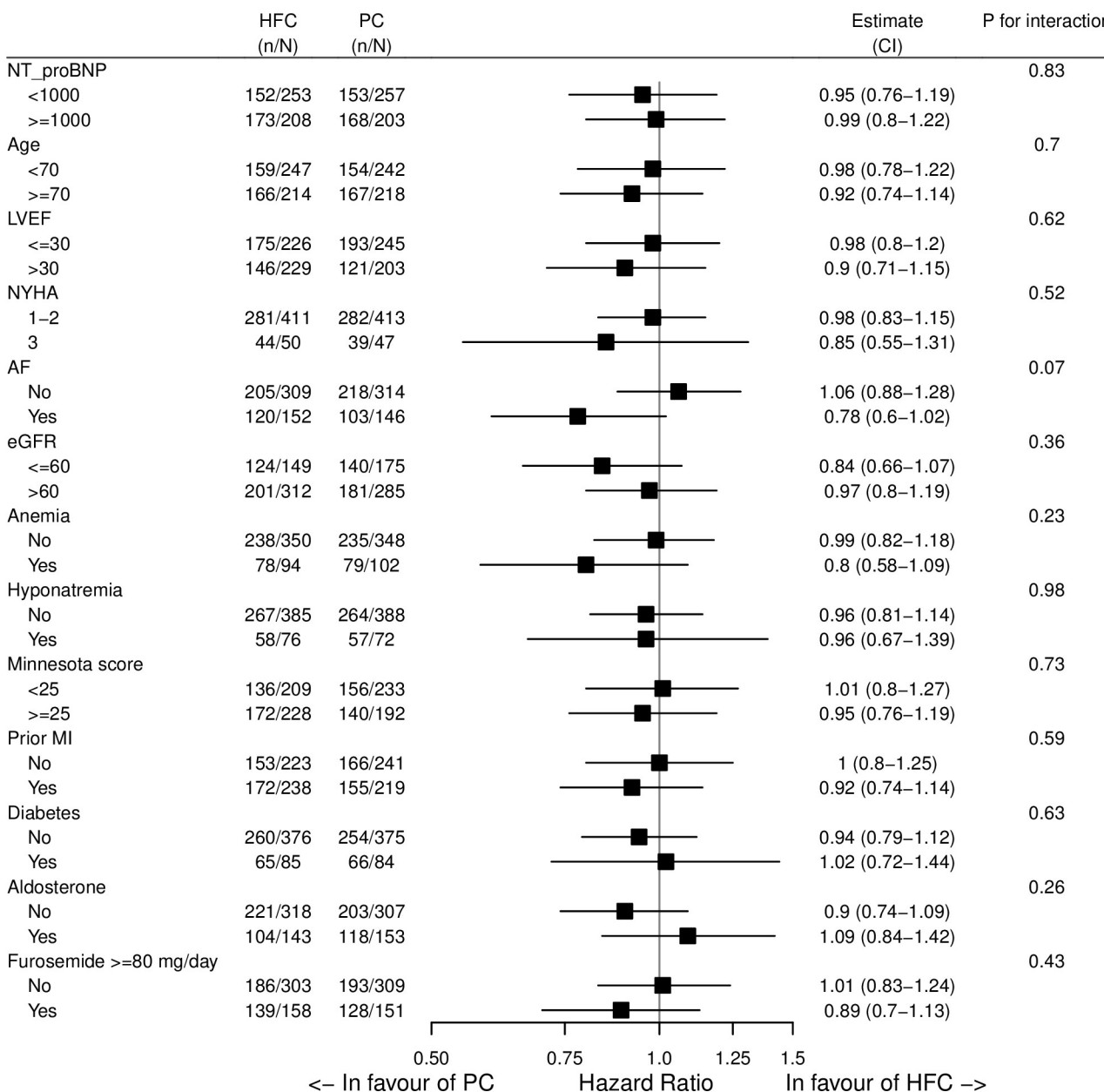

| | HFC (n/N) | PC (n/N) | | Estimate (CI) | P for interaction |
|---|---|---|---|---|---|
| NT_proBNP | | | | | 0.83 |
| <1000 | 152/253 | 153/257 | | 0.95 (0.76–1.19) | |
| >=1000 | 173/208 | 168/203 | | 0.99 (0.8–1.22) | |
| Age | | | | | 0.7 |
| <70 | 159/247 | 154/242 | | 0.98 (0.78–1.22) | |
| >=70 | 166/214 | 167/218 | | 0.92 (0.74–1.14) | |
| LVEF | | | | | 0.62 |
| <=30 | 175/226 | 193/245 | | 0.98 (0.8–1.2) | |
| >30 | 146/229 | 121/203 | | 0.9 (0.71–1.15) | |
| NYHA | | | | | 0.52 |
| 1–2 | 281/411 | 282/413 | | 0.98 (0.83–1.15) | |
| 3 | 44/50 | 39/47 | | 0.85 (0.55–1.31) | |
| AF | | | | | 0.07 |
| No | 205/309 | 218/314 | | 1.06 (0.88–1.28) | |
| Yes | 120/152 | 103/146 | | 0.78 (0.6–1.02) | |
| eGFR | | | | | 0.36 |
| <=60 | 124/149 | 140/175 | | 0.84 (0.66–1.07) | |
| >60 | 201/312 | 181/285 | | 0.97 (0.8–1.19) | |
| Anemia | | | | | 0.23 |
| No | 238/350 | 235/348 | | 0.99 (0.82–1.18) | |
| Yes | 78/94 | 79/102 | | 0.8 (0.58–1.09) | |
| Hyponatremia | | | | | 0.98 |
| No | 267/385 | 264/388 | | 0.96 (0.81–1.14) | |
| Yes | 58/76 | 57/72 | | 0.96 (0.67–1.39) | |
| Minnesota score | | | | | 0.73 |
| <25 | 136/209 | 156/233 | | 1.01 (0.8–1.27) | |
| >=25 | 172/228 | 140/192 | | 0.95 (0.76–1.19) | |
| Prior MI | | | | | 0.59 |
| No | 153/223 | 166/241 | | 1 (0.8–1.25) | |
| Yes | 172/238 | 155/219 | | 0.92 (0.74–1.14) | |
| Diabetes | | | | | 0.63 |
| No | 260/376 | 254/375 | | 0.94 (0.79–1.12) | |
| Yes | 65/85 | 66/84 | | 1.02 (0.72–1.44) | |
| Aldosterone | | | | | 0.26 |
| No | 221/318 | 203/307 | | 0.9 (0.74–1.09) | |
| Yes | 104/143 | 118/153 | | 1.09 (0.84–1.42) | |
| Furosemide >=80 mg/day | | | | | 0.43 |
| No | 186/303 | 193/309 | | 1.01 (0.83–1.24) | |
| Yes | 139/158 | 128/151 | | 0.89 (0.7–1.13) | |

<– In favour of PC    Hazard Ratio    In favour of HFC –>

**Fig 2. Occurrence of cardiovascular death or discharge for heart failure in subgroups.** Abbreviations: PC = primary care, HFC = heart failure clinic, n = the number of events of the primary outcome, N = the number of patients in the PC or HFC group, CI = confidence interval, NT_proBNP = N-terminal Pro-B-Type Natriuretic Peptide, LVEF = Left ventricular ejection fraction, NYHA = New York Heart Association functional classification, AF = atrial fibrillation, eGFR = estimated glomerular filtration rate, MI = myocardial infarction.

and HFC groups (*S2 Fig* and *S2 Table*). In the interaction analyses, no differences were observed between the PC and HFC groups in subgroups of the number of HF hospitalization regardless of the outcome of interest (*S2 Table*). However, since the numbers of both patients and outcomes were low, particularly in the groups of patients with prior HF hospitalizations within the first 2 years following randomization, these results should be interpreted with caution.

### Coverage and drug adherence in 5-year survivors

In 5-year survivors, a low proportion had not been covered by RASi (HFC: 3.0%; PC: 2.2%) and BB (HFC: 2.1%; PC: 3.1%) during the 5 years of follow-up, and among individuals covered with RASi or BB the PDC-level for both drugs was high (*Fig 3*). In contrast, approximately half of the patients had not been covered by MRA (HFC: 51.9%; PC: 52.0%). We observed no difference in means of PDC between the HFC and the PC group (RASi: p = 0.78; BB: p = 0.74; MRA: p = 0.47).

## Discussion

### Main findings

We observed that HFrEF patients on optimal medical therapy did not benefit from additional years of extended follow-up in a specialized HF clinic. The risk of relevant HF endpoints and adherence to neurohormonal blockade did not differ between patients followed in an HFC

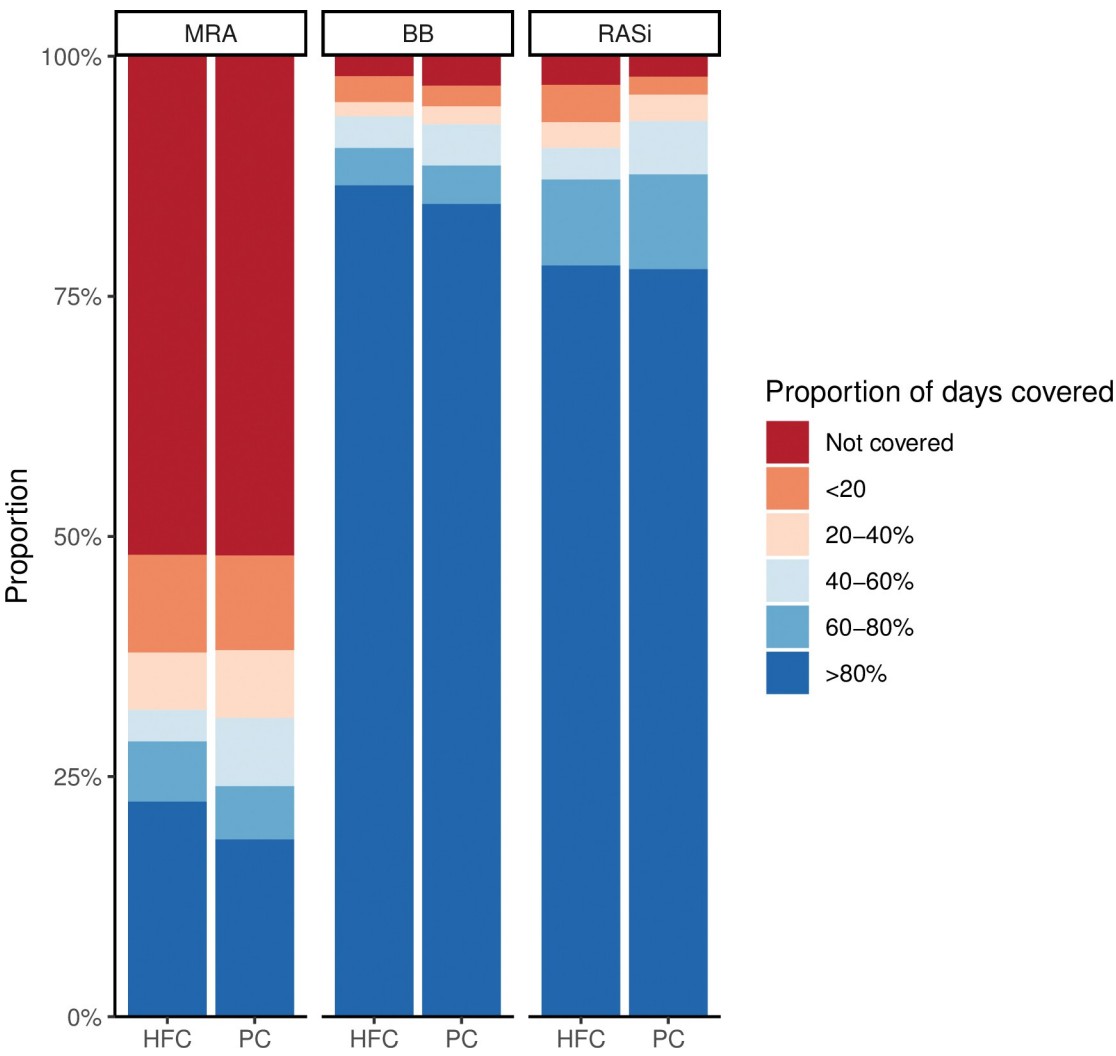

**Fig 3. Adherence to mineralocorticoid receptor antagonists, beta-blockers, and renin-angiotensin system inhibitors in 5-year survivors during the first 5 years of follow-up.** Abbreviations: MRA = mineralocorticoid receptor antagonists, BB = beta blocker, RASi = renin-angiotensin receptor system inhibitors, PC = primary care, HFC = heart failure clinic.

compared to PC. Our results underscore the need for the development of improved monitoring programs to improve outcomes for patients receiving guideline-directed therapy.

## Other studies and subgroups

In contrast to our study, two recent studies using observational data from the Swedish HF Registry observed a lower mortality rate and a higher rate of HF hospitalizations in the HFC compared to the PC group [13, 14]. However, their results may have been affected by residual confounding and imputed data. Secondly, the Swedish HF Registry uses a broad inclusion criterion, and it is not fully clear whether all patients were included following fully optimized medical treatment in an HFC, both factors limiting the comparison to our study. According to our knowledge, no previous trials have evaluated 10 years of follow-up in two different parts of a public health care system in a randomized clinical trial setting. We did not observe any additional effect of follow-up in a specialized HF clinic. This may be explained by several factors. First, there is a possibility of selection bias at enrolment. Patients included in a randomized clinical trial are generally healthier than patients in general [15], and it cannot be ruled out that a sicker population would have benefitted from continued follow-up. Second, the intervention was too weak considering that the patients were already receiving optimal medical therapy before randomization leaving only little room for further optimization. Third, cross over between the two arms. Within the original study period, only minor cross-over occurred [2] but during a 10 years period with a high proportion of HF re-hospitalizations, it may have occurred over time. Finally, no statistical interactions between treatment allocation and traditional subgroups on the risk of the primary outcome were observed, supporting the overall neutral result. Therefore, more sophisticated risk stratification than the identification of subgroups is probably needed in the future to identify patients that need specialized care for years [16].

## Adherence to guideline therapy

In patients who were alive after 5 years, we evaluated the 5-year adherence to HF therapy. Here we did not observe any difference between groups. Considering that specialized HF clinics are designed to improve adherence, this finding was surprising. However, it supports the overall neutral result considering the lack of effect of patient education *per se* observed in the COACH trial [17]. It may be further explained by the fact that the patients were already on optimal medical therapy and all patients were educated in self-care and disease management. It should be noted that adherence to MRAs was poor in both groups. MRAs improve life expectancy considerably [18, 19] and more focus on adherence to MRA is needed in future studies together with the implementation of sacubitril-valsartan [20] and sodium-glucose inhibitors [21, 22]. Somewhat surprisingly, the proportion of patients with a PDC level>80% appeared to be slightly higher BB compared to RASi. However, it should be mentioned that the patients in the present study were made sure to tolerate the HF medication to the max tolerable dose before randomization, which may have ensured relatively high adherence to BB. Additionally, medications that are prescribed more than once daily complicate the calculation of PDC. Thus, calculations of PDC and differences in adherence between drugs should be interpreted with caution.

## Clinical perspectives

We observed that although the patients were on optimal medical treatment, the mortality risk and HF hospitalization remained high for both patients followed in the PC and specialized HF clinics. Therefore, it may be suggested that improvement in monitoring HFrEF patients on already optimal medical therapy is needed. Follow-up by highly educated doctors and nurses

who only treat HF patients compared to follow-up by nurses and doctors in primary care who treat several diseases daily did not result in an improvement in HF outcomes. Whether e.g the use of a CardioMems system can improve outcomes in less symptomatic patients is under investigation [23, 24]. Within the last years, new drugs have been developed–Sacubitril-Valsartan [20] and Sodium-dependent glucose cotransporters [21]–and implementation of these is also needed to improve clinical outcomes for HFrEF patients. Patients followed in the PC should be referred to specialized HF clinics to initiate these treatments and during a re-admission for worsening HF cardiologists should be ready to initiate new treatments to avoid delay in the initiation of new drug discoveries when endorsed into clinical guidelines [25–28].

## Methodological considerations

The strengths of this multicenter, open-labeled sub-study of 921 patients include a long and nearly complete follow-up period and a large number of events ensured by the comprehensive Danish registries. However, important limitations need to be addressed. First, patients were selected using a list of criteria, including HFrEF patients as well as excluding patients with NYHA-class 4, malignant disease, and an expected lifetime of < 5 years. Although the excluded group of patients may have had a larger number of events, this does not contradict our conclusion. However, the generalizability is limited to NYHA-class 1–3 HFrEF patients without recent malignant disease in a universal healthcare system such as Denmark. Additionally, the majority of the Danish population is Caucasian, thus our results may not be generalizable to non-Caucasian individuals. Secondly, cross-over after the follow-up end of the Northstar trial is a possibility, particularly given the numerous HF hospitalizations across the total follow-up period. However, for patients followed in the PC, we observed no gap in adherence to evidence-based medical treatment nor in survival during the first 5 years after follow-up. Furthermore, the results of this study depend on the quality of data including the discharge diagnosis. In this regard, it is important to note that the discharge diagnosis of HF has a low sensitivity [10], which could have underestimated the risk of HF hospitalization in both groups. Lastly, although type 2 errors cannot be excluded, we consider it unlikely given the robustness of our results in both primary and secondary outcomes.

## Conclusions

HFrEF patients on optimal medical therapy did not benefit from extended follow-up in a specialized HF clinic. The risk of HF hospitalization and cardiovascular death, and adherence to the neurohormonal blockade, did not differ between patients followed in an HFC compared to PC. Given the high HF hospitalization and mortality rates, our results underscore the need for the development of improved monitoring programs including new technologies to improve outcomes for patients on guideline-directed therapy.

## Clinical perspectives

**Competency in medical knowledge.**   We observed that although the patients were on optimal medical treatment, the mortality risk and HF hospitalization remained high for both patients followed in the PC and specialized HF clinics. Therefore, it may be suggested that improvement in monitoring HFrEF patients on already optimal medical therapy is needed. Follow-up by highly educated doctors and nurses who only treat HF patients compared to follow-up by nurses and doctors in primary care who treat several diseases daily did not result in an improvement in HF outcomes.

**Translational outlook.**   Whether e.g the use of a CardioMems system can improve outcomes in less symptomatic patients is under investigation [23, 24]. Within the last years, new

drugs have been developed–Sacubitril-Valsartan [20] and Sodium-dependent glucose cotransporters [21]–and implementation of these is also needed to improve clinical outcomes for HFrEF patients.

## Supporting information

**S1 Methods. Inclusion and exclusion criteria.**
(PDF)

**S1 Table. Three most common primary causes of rehospitalizations (overnight stays only) during follow-up according to the type of disease.**
(PDF)

**S2 Table. The 8-year hazard rate of all-cause death, cardiovascular death, cardiovascular death or heart failure, and heart failure among 2-year survivors in groups defined by the number of prior hospitalizations for heart failure within the first two years following randomization.**
(PDF)

**S1 Fig. Flowchart.**
(PDF)

**S2 Fig. The 8-year absolute risk of all-cause death, cardiovascular death, discharge for heart failure, and cardiovascular death or discharge for heart failure in 2-year survivors in groups defined by the number of prior discharge diagnoses for heart failure.**
(PDF)

## Acknowledgments

The technical assistance from the staff at the Department of Clinical Chemistry, Frederiksberg University Hospital is deeply acknowledged. M. Malmborg and M. Schou had full access to all the data in the study and takes responsibility for the integrity of the data and the accuracy of the data analysis.

## Author Contributions

**Conceptualization:** Morten Malmborg, Lars Kober, Christian Torp-Pedersen, Jawad H. Butt, Deewa Zahir, Christian D. Tuxen, Mikael K. Poulsen, Emil Fosbol, Per Hildebrandt, Charlotte Andersson, Finn Gustafsson, Morten Schou.

**Data curation:** Morten Schou.

**Formal analysis:** Morten Malmborg.

**Funding acquisition:** Morten Schou.

**Investigation:** Morten Malmborg, Ali Assad Turky Al-Kahwa, Morten Schou.

**Methodology:** Morten Malmborg, Lars Kober, Christian Torp-Pedersen, Morten Schou.

**Project administration:** Morten Schou.

**Supervision:** Morten Schou.

**Visualization:** Morten Malmborg, Morten Schou.

**Writing – original draft:** Morten Malmborg, Morten Schou.

**Writing – review & editing:** Morten Malmborg, Ali Assad Turky Al-Kahwa, Lars Kober, Christian Torp-Pedersen, Jawad H. Butt, Deewa Zahir, Christian D. Tuxen, Mikael K. Poulsen, Christian Madelaire, Emil Fosbol, Gunnar Gislason, Per Hildebrandt, Charlotte Andersson, Finn Gustafsson, Morten Schou.

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
