## [Decision Letter · Decision Letter 0]

22 Nov 2022

PONE-D-22-18899Specialized heart failure clinics versus primary care: Extended registry-based follow-up of the NorthStar TrialPLOS ONE

Dear Dr. Malmborg,

Thank you for submitting your manuscript to PLOS ONE. After careful consideration, we feel that it has merit but does not fully meet PLOS ONE’s publication criteria as it currently stands. Therefore, we invite you to submit a revised version of the manuscript that addresses the points raised during the review process.

We look forward to receiving your revised manuscript.

Kind regards,

Giuseppe Gargiulo, MD, PhD

Academic Editor

PLOS ONE

Journal Requirements:

"This work was funded by the Danish Heart Foundation (Grant number: 18-R121-A8218). "

Additional Editor Comments:

This is an interesting study.

It would be interesting to account for additional events during follow-up. Please include an additional analysis comparing groups based on number of HF re-hospitalizations.

Please check figure 1, the titles on the KM curves seem not corresponding to the figure caption.

Please better comment on the borderline P for interaction for AF in Fig 2. Statistical significance could be missing simply due to power issues.

Can the authors provide more information on the reasons for re-hospitalizations during the follow-up?

The rates of device therapy at randomization is very low. It would be important to have info during follow-up.

Please add more details on the initial trial. The paper must be self-standing and the readers/reviewers should not go searching for other info from different prior publications.

Reviewers' comments:

Reviewer's Responses to Questions

**Comments to the Author**

1. Is the manuscript technically sound, and do the data support the conclusions?

Reviewer #1: Yes

Reviewer #2: Yes

Reviewer #3: Yes

2. Has the statistical analysis been performed appropriately and rigorously? 

Reviewer #1: Yes

Reviewer #2: Yes

Reviewer #3: Yes

3. Have the authors made all data underlying the findings in their manuscript fully available?

Reviewer #1: No

Reviewer #2: No

Reviewer #3: No

4. Is the manuscript presented in an intelligible fashion and written in standard English?

Reviewer #1: Yes

Reviewer #2: Yes

Reviewer #3: Yes

5. Review Comments to the Author

Reviewer #1: PONE-D-22-18899: statistical review

This study compares long-term survival between heart failure patients who have been randomly assigned to follow-up in a specialized HF clinic or primary care. 5-year adherence to prescribed neurohormonal blockade in 5-year survivors is also assessed. Traditional survival analysis methods are correctly exploited to compare the survival trajectories between the two groups and beta regression is correctly considered for comparing adherence proportions. I have nothing to say about this paper: the research question is well posed, statistical methods are appropriate, results are correctly interpreted and possible limitations of the study are nicely remarked in the discussion.

Reviewer #2: In this manuscript, authors report findings from the extended follow-up of a trial conducted in Denmark (NorthStar) that randomized patients with heart failure with reduced ejection fraction (HFrEF) to be followed up in primary care or in a specialized HF clinic once treatment had been optimized.

FU data were obtained using linked electronic health records.

Overall, the manuscript is clear and easy to read till the end.

However, I think that the main limitation of this study, as acknowledged by the authors, is that during a period of up to 10 years many bad things can happen to patients with heart failure, and a substantial proportion of those randomized to a primary care clinic follow-up would have received specialist input due to clinical deterioration which, in my opinion, limits the relevance of these findings.

Authors might add a comment to adherence to beta-blockers, which seems higher that that to RASi.

Reviewer #3: The authors present a very interesting study on the value of specialized heart failure clinics versus primary care. It is a relevant study, but I have some comments:

- Although an NT-proBNP > 1000 pg/mL might identify high-risk patients, an HF hospitalization (43% in the initial study) the previous year might be a marker of higher risk than NTproBNP values.

- The authors use both MRA and ARA. ARA is not defined in the Abbreviation nor Table 1. Interestingly, the use of MRA appears to increase from randomization (33%) to the 5-year follow-up MRA (HFC: 51.9%; PC: 52.0%). It may be associated with the publication of pivotal MRA trials in HF. However, it is surprising that PC prescribed MRA at a relatively high percentage.

- Was CRT or ICD different at 5-year follow-up? The use at baseline is surprisingly low.

6. PLOS authors have the option to publish the peer review history of their article (what does this mean?). If published, this will include your full peer review and any attached files.

Reviewer #1: No

Reviewer #2: No

Reviewer #3: No

---

## [Author Response · Author response to Decision Letter 0]

18 Feb 2023

Please see the attached file "Response to reviewers".

---

## [Decision Letter · Decision Letter 1]

6 Mar 2023

PONE-D-22-18899R1Specialized heart failure clinics versus primary care: Extended registry-based follow-up of the NorthStar TrialPLOS ONE

Dear Dr. Malmborg,

Thank you for submitting your manuscript to PLOS ONE. After careful consideration, we feel that it has merit but does not fully meet PLOS ONE’s publication criteria as it currently stands. Therefore, we invite you to submit a revised version of the manuscript that addresses the points raised during the review process.

We look forward to receiving your revised manuscript.

Kind regards,

Giuseppe Gargiulo, MD, PhD

Academic Editor

PLOS ONE

Additional Editor Comments:

The authors provided a properly revised version of the manuscript with almost all points addressed.

There is still 1 of my previous points to be addressed, specifically point 1.4 in which I did not ask for specific new analyses but simply a table or text reporting reasons of re-hospitalizations during follow-up (new acute coronary syndrome, infections, diuretic resistance, worsening of renal function etc.).

Also few points from reviewer 4 should be addressed.

Reviewers' comments:

Reviewer's Responses to Questions

**Comments to the Author**

1. If the authors have adequately addressed your comments raised in a previous round of review and you feel that this manuscript is now acceptable for publication, you may indicate that here to bypass the “Comments to the Author” section, enter your conflict of interest statement in the “Confidential to Editor” section, and submit your "Accept" recommendation.

Reviewer #2: All comments have been addressed

Reviewer #3: All comments have been addressed

Reviewer #4: All comments have been addressed

2. Is the manuscript technically sound, and do the data support the conclusions?

Reviewer #2: Partly

Reviewer #3: Yes

Reviewer #4: Yes

3. Has the statistical analysis been performed appropriately and rigorously? 

Reviewer #2: Yes

Reviewer #3: Yes

Reviewer #4: Yes

4. Have the authors made all data underlying the findings in their manuscript fully available?

Reviewer #2: Yes

Reviewer #3: No

Reviewer #4: Yes

5. Is the manuscript presented in an intelligible fashion and written in standard English?

Reviewer #2: Yes

Reviewer #3: Yes

Reviewer #4: Yes

6. Review Comments to the Author

Reviewer #2: Authors performed additional analyses but I still think that the main limitation of this study remains a (very likely) high rate of cross over: as acknowledged, a substantial proportion of those randomized to a primary care clinic follow-up had been admitted with heart failure and would have received specialist input due to clinical deterioration. This, in my opinion, limits

the relevance of these findings that could be, eventually, summarised in a much shorter research letter.

Reviewer #3: All comments have been addressed by the authors. I believe the manuscript can be published in the current form.

Reviewer #4: In this manuscript, Dr. Malmborg and colleagues tried to assess if there is a difference in hospitalization for heart failure or cardiovascular mortality comparing specialized heart failure clinics versus primary care through an extended post-hoc registry-based follow-up study of a randomized controlled trial.

Overall, this is a quite nicely written article, greatly improved already by previous revisions, with a clinical relevance.

However, some relevant issues need to be addressed by the authors:

- Albeit the study was conducted rigorously, the result is in contrast to recent results from registry studies involving far larger numbers of patients (https://onlinelibrary.wiley.com/doi/full/10.1002/ehf2.13848; https://www.internationaljournalofcardiology.com/article/S0167-5273(21)01334-6/fulltext). This should be expanded upon in the discussion section, pointing out why there may be this discrepancy.

- Table 1: The authors report an haemoglobin mean level of 8.6, but only about 20% of patients with anemia. How is this possible? What are the upper letter a in anemia and upper letter b in hyponatremia for?

- Figure 2: On the bottom near the hazard ratio should be specified whether left or right is in favour of HFC or PC

- Please note that the text requires a careful proofreading, since there are several grammar lapses

7. PLOS authors have the option to publish the peer review history of their article (what does this mean?). If published, this will include your full peer review and any attached files.

Reviewer #2: No

Reviewer #3: No

Reviewer #4: **Yes: **Christian Basile

---

## [Author Response · Author response to Decision Letter 1]

27 Apr 2023

Dear Giuseppe Gargiulo and reviewers

The current study has previously been under review twice by Plos One and was ultimately encouraged to be resubmitted with positive reviews and minor comments. We thank you for your previous constructive criticism of our manuscript. The manuscript has undergone a revision according to the suggestions by the reviewers, and we think the manuscript has been improved, and we kindly ask you to consider the manuscript for publication in Plos One. We have addressed the comments by the editor and reviewers, and we hope that with these changes, the manuscript will be acceptable for consideration in Plos One. 

Yours sincerely,

Morten Malmborg, MD

Corresponding author

---

## [Decision Letter · Decision Letter 2]

15 May 2023

Specialized heart failure clinics versus primary care: Extended registry-based follow-up of the NorthStar Trial

PONE-D-22-18899R2

Dear Dr. Malmborg,

We’re pleased to inform you that your manuscript has been judged scientifically suitable for publication and will be formally accepted for publication once it meets all outstanding technical requirements.

Kind regards,

Giuseppe Gargiulo, MD, PhD

Academic Editor

PLOS ONE

Additional Editor Comments (optional):

All comments were addressed. The manuscript is improved and can now be accepted for publication.

Reviewers' comments:

Reviewer's Responses to Questions

**Comments to the Author**

1. If the authors have adequately addressed your comments raised in a previous round of review and you feel that this manuscript is now acceptable for publication, you may indicate that here to bypass the “Comments to the Author” section, enter your conflict of interest statement in the “Confidential to Editor” section, and submit your "Accept" recommendation.

Reviewer #4: All comments have been addressed

2. Is the manuscript technically sound, and do the data support the conclusions?

Reviewer #4: Yes

3. Has the statistical analysis been performed appropriately and rigorously? 

Reviewer #4: Yes

4. Have the authors made all data underlying the findings in their manuscript fully available?

Reviewer #4: Yes

5. Is the manuscript presented in an intelligible fashion and written in standard English?

Reviewer #4: Yes

6. Review Comments to the Author

Reviewer #4: The manuscript was greatly improved, the images are now clearer and as it is I have no further comment.

7. PLOS authors have the option to publish the peer review history of their article (what does this mean?). If published, this will include your full peer review and any attached files.

Reviewer #4: **Yes: **Basile Christian

---

## [Editor Report · Acceptance letter]

31 May 2023

PONE-D-22-18899R2 

Specialized heart failure clinics versus primary care: Extended registry-based follow-up of the NorthStar Trial 

Dear Dr. Malmborg:

I'm pleased to inform you that your manuscript has been deemed suitable for publication in PLOS ONE. Congratulations! Your manuscript is now with our production department. 

Kind regards, 

on behalf of

Dr. Giuseppe Gargiulo 

Academic Editor

PLOS ONE